# Measurement and Prediction of Railway Noise Case Study from Slovakia

**DOI:** 10.3390/ijerph17103616

**Published:** 2020-05-21

**Authors:** Miroslav Němec, Anna Danihelová, Tomáš Gergeľ, Miloš Gejdoš, Vojtěch Ondrejka, Zuzana Danihelová

**Affiliations:** 1Faculty of Wood Sciences and Technology, Department of Physics, Electrical Engineering and Applied Mechanics, Technical University in Zvolen, T.G Masaryka 24, 96001 Zvolen, Slovakia; nemec@tuzvo.sk; 2Faculty of Wood Sciences and Technology, Department of Fire Protection, Technical University in Zvolen, T.G Masaryka 24, 96001 Zvolen, Slovakia; danihelova@acoustics.sk; 3National Forest Centre, Forest Research Institute, 96001 Zvolen, Slovakia; tomas.gergel@nlcsk.org (T.G.); ondrejka.vojtech@gmail.com (V.O.); 4Faculty of Forestry, Department of Forest Harvesting, Logistics and Amelioration, Technical University in Zvolen, T. G. Masaryka 24, 96001 Zvolen, Slovakia; 5Institute of Foreign Languages, Technical University in Zvolen, T. G. Masaryka 24, 96053 Zvolen, Slovakia; zuzana.danihelova@tuzvo.sk

**Keywords:** noise, prediction, rail transport, noise maps, CadnaA

## Abstract

The paper deals with comparing the measurement of noise from the railroads in the residential zone of the town of Zvolen with the results calculated using the prediction methods “Schall 03“ (Deutsche Bundesbahn, 1990) and “Methodical instructions for the calculation of sound pressure level from transport” (MPVHD). The first is used in the Slovakia and second in the Czech Republic. The measurement results and the results obtained from the prediction methods for both measurement locations were evaluated graphically and statistically. The evaluation of the conformity of the measurement with the prediction showed that the results obtained using the method “Schall 03” are in better agreement with the measurement.

## 1. Introduction

The objective of the present study is to compare the in situ measurement results of the noise from the railway transport in the central Europe area with the results obtained from the prediction methods “Schall 03“ and “Methodical instructions for the calculation of sound pressure level from transport” (MPVHD). On the basis of this comparison, we aim to determine the more appropriate prediction method and to prove the justification of using national corrections for wagon stock specificities in the territory of the Slovak Republic. With increasing traffic in big cities, it will be important in the future to verify older noise predictive methods, whether they are still taking into account reality or need to be revised, respectively, or if developing new methods is necessary. The results of this work should contribute to this.

According to the information on mapping noise in the member states collected by the European Environmental Agency (EEA) in 2010, noise from railway transport during the day affects approximately 12 million inhabitants of the European Union when the sound pressure level is higher than 55 dB, and approximately 9 million inhabitants are affected by the noise at night, with the sound pressure level being higher than 50 dB. However, these data are higher without any doubt since the EAA initiative to map noise is focused on urban agglomerations with more than 250,000 inhabitants and on railways with more than 60,000 trains per year. The railway noise issue is concentrated in Central Europe, where the majority of inhabitants live and the amount of goods trains is the highest [1,2,3,4,5].

In the European Union, noise emissions are discussed within the Directive 2002/49/EC of the European Parliament and of the Council [6]. This directive is focused on determining the common approach to preventing the harmful effects of environmental noise. Its objective is to provide a joint background for solving the issue of noise in the entire EU. These legal regulations are in the Slovak Republic implemented in the form of Act no. 2/2005 Coll., as amended by later regulations on Assessment and Control of Environmental Noise [7]. The problem with analyzing the noise exposure is that the individual member states use different measurement and noise prediction methods, and the maximum allowable values of the determining quantities for the road and railway transport are different [8,9].

In 2012, the EAA assessed the noise exposure of the EU state inhabitants in 467 urban agglomerations (roads, railways, airports, and industrial sources of noise) at 86 larger airports and on 186,600 km of major roads and 44,320 km of major railways outside the urban agglomerations. Figure 1 illustrates the individual results in and outside the urban agglomerations according to the combined noise indicator *L*_den_ value above 55 dB because the noise higher than 55 dB is considered noise pollution. The combined noise indicator *L*_den_ allows us to evaluate the noise emissions in relation to people annoyance of noise during 24 h. This indicator is widely used for exposure to noise assessment in health effect studies.

According to European Environment Agency (EEA) data, Slovakia is second of the eight states in Europe that are most affected by railway noise, according to the share of their population, with *L*_den_ more than 55 dB when more than 9% of Slovakia’s population is exposed to noise from rail transport.

In addition, this study [6] proved that road transport within and outside urban agglomerations is the dominant source of noise. Furthermore, a significant contributor to the overall noise burden is rail transport [10,11,12]. Eighteen million people (approx. 10 million in urban agglomerations and 8 million outside them) are exposed to sound pressure levels higher than the indicator *L*_den_ 55 dB mentioned in the Directive of the European Parliament and of the Council 2002/49/EC. Environmental noise causes 15,900 premature deaths annually. This report summarises only selected urban agglomerations and places; therefore, the overall effects of noise on human health will be even more significant.

## 2. Materials and Methods

### 2.1. Measurement Methodology

Measuring the noise in the environment is administered by the standard ISO 1996-2:2017 “Acoustics—description, measurement, and assessment of environmental noise—Part 2: Determination of sound pressure levels” [13]. The present measurements were carried out using a hand-held sound pressure analyzer, Brüel& Kjær 2270™. It is an analyzer working real-time in accuracy Class I. Its dynamic range is 120 dB [14].

Following the objective of the study, a suitable measuring location was selected (Figure 2). It is located in the central part of the town of Zvolen at a busy railway connecting significant cities of the region–Zvolen and Banská Bytsrica. This region is the third-largest one in Slovakia regarding the number of inhabitants, which also corresponds to the overall traffic intensity. The selected location is situated close to the hospital. The residual noise from the surrounding sources (mainly road transport) is during the measurement sufficiently decreased; this was also proven during the actual measurement campaigns. The measurements and predictions were carried out repeatedly (on 3 October 2011 measurement campaign ZV1 and on 24 April 2015 measurement campaign ZV2) because, since 17 November 2014, following the Directive of the Government of the Slovak Republic, the Railways of the Slovak Republic provide free transport for certain population groups (students, retired).

Both measurement campaigns were carried out during the reference time “day” (12 h) from 6 a.m. to 6 p.m. After the calibration, the sound pressure analyzer was placed at the measurement location on a tripod, 7.5 m far from the axis of railway no. 170, at the height of 1.5 m above the surrounding terrain, perpendicular to the railway line axis (Figure 3). The overall height of the measurement location was 3.5 m above the railway line. The embankment is 0.5 m high. The number, type, and direction of the trains passing by were recorded continuously. The course of sound exposure was recorded continuously during the reference time “day” (12 h) using the sound pressure analyzer with the corresponding sound record. The equivalent sound level pressure A *L*_Aeq,12h_ from the railway transport for the reference time “day” at the given measurement location was calculated according to the ISO 1996-2:2017 [13].

### 2.2. Prediction Methods

For comparing the objectified measured values of the railway transport sound pressure, various prediction methods are used worldwide [9,15,16,17,18,19]. The present study considers two selected methods used in the Slovak Republic and in the Czech Republic, the Schall 03 method [20] and MPVHD. The Deutsche Bundesbahn developed and uses the Schall 03 2006 prediction method. It is used for predicting railway noise and creating railway noise maps in Slovakia. The equivalent sound pressure level A is, in this case, called the emission level *L_m,E_* and is determined at the perpendicular distance of 25 m from the railway line axis at the height of 4 m above the terrain. Emission levels *L_m,E_* are determined according to the equation
(1)$$L_{m,E}=10\cdot \log\left[\sum 10^{\left(0,1\cdot (51+D_{Fz}+D_{D}+D_{L}+D_{V})\right)}\right]+D_{Tt}+D_{Br}+D_{Lc}+D_{Ra}$$
where *D_Fz_, D_D_, D_L_, D_s_* are corrections for the specific train and *D_Tt_, D_Br_, D_Lc_, D_Ra_* are corrections characterizing the railway line. A more detailed explanation of the methodology can be found in [20].

The prediction method MPVHD [21] has been used in the Czech Republic since 1977. The latest legislative version of the valid prediction procedures for measuring the noise of ground transport is the methodical instructions for the calculation of sound pressure level from transport, whose author is Liberko. These instructions were issued in 1991 and amended in 1996.

The equivalent sound pressure level A in dB is established for the assessed site located at a distance of 7.5 m from the railway line axis. The equation is as follows:(2)Y=10.logX+40

The value X (in dB as well) is calculated according to the equation:(3)$$X=140.^{}F_{4}.^{}F_{5}.^{}F_{6}.^{}m$$
where *F*_4_ is the factor characterizing the impact of traction, *F*_5_ is the factor of instantaneous speed at the given point of the railway line, and *F*_6_ is the factor considering the average total number of vehicles. The methodology is further explained in Liberko (1996).

For assessing the differences of the determining quantities between the prediction and measurement, Bland–Altman plots were used. These plots compare two data sets (measurement and the given prediction) using the difference between them. This difference is the function of the averaged differences between the two sets [22]. The overall noise situation in the vicinity of the measurement location is illustrated using the noise maps, which were created using the CadnaA program [23,24].

## 3. Results and Discussion

At the time of the first measurement campaign, the weather conditions were suitable for carrying out the measurement. The information about the composition and number of individual train types are provided in Table 1.

Although the railway line is electrified, only 27% of the trains use the electric traction system. It is caused partially by the fact that the electrified line does not continue past Banská Bystrica, and therefore, the regional trains between Zvolen and Banská Bystrica are often engine-driven ordinary passenger trains. The equivalent sound pressure level A in the reference time “day” for this measurement was *L*_Aeq,12h_ = 59.3 dB. The values of the determining quantities (equivalent sound pressure level A–*L*_Aeq,1h_) acquired from the prediction using the Schall 03 method and MPVHD were compared with the values acquired from the measurements at the measurement location M1 Zvolen. The values were compared graphically and statistically. Figure 4 illustrates the graphical evaluation of the results acquired from both prediction methods and from measurement in one-hour time intervals.

The connection of individual equivalent sound levels A in the given intervals is for the sake of easier orientation and interpretation of differences between individual methods. Figure 3 shows that the tendency of the determining quantities acquired by the prediction methods and by the measurement is similar. However, it is apparent that the values of equivalent sound levels A in corresponding time intervals are different. The biggest difference in *L*_Aeq,1h_ was for the interval of 10 a.m. and 11 a.m., when only one train passing by was recorded. The prediction methods could not take into consideration this fact correctly.

The *L_Aeq,1h_* levels obtained by the Schall 03 prediction method and by the measurement are compared using Bland–Altman plots [22] in Figure 5. Figure 6 presents the comparison of *L_Aeq,1h_* levels obtained by the prediction method MPVHD and by the measurement. In these plots, the central horizontal line (thin line) represents the mean difference between the *L_Aeq,1h_* levels, and the boundary lines (thick lines) are defined as the difference mean ± doubled standard deviation (SD) of the differences of averages. The measured values of *L_Aeq,1h_* levels (in dB) are illustrated by the red line in both plots. The vertical axis illustrates the differences between the measured and predicted values. From these plots, the following can be seen: the mean differences between the measured and predicted values obtained by the Schall 03 method are smaller than in the case of MPVHD method. The use of the Schall 03 method provided more values that are only minimally different from the measured values. In the case of prediction using both methods (MPVHD and Schall 03), the equivalent sound levels A are systematically overestimated. Nevertheless, the values obtained by the Schall 03 method are different only to a smaller extent regarding the measured values. Following the above-mentioned facts, it can be stated that the prediction method Schall 03 provides more actual values of equivalent sound levels A *L_Aeq,1h_*.

The repeated measurement was carried out right after the decision of the Government of the Slovak Republic to establish free transport for selected population groups (students, retired) provided by the national railway company. During this measurement, the intensity of the railway transport increased by approx. 20% on this part of the line. It was proven by 50 recorded trains passing by during the reference time “day”. Information about the composition and number of individual train types are provided in Table 2.

The share of electric traction decreased to 10%. It is apparent that the electrification potential of this part of the line is used at the minimum. The equivalent sound pressure level A in the reference time “day” was this time *L*_Aeq,12h_ = 61.5 dB.

Figure 7 provides a comparison of the values of the determining quantities (equivalent sound pressure level A–*L*_Aeq,1h_) obtained from the measurement with the values provided by the prediction method in one-hour intervals. The results of *L*_Aeq,__1h_ obtained by the prediction method Schall 03 are almost identical to the measurement. A more significant difference was recorded between 11 a.m. and 12 a.m. when the longest goods train was passing along the given part of the railway line. The MPVHD method also records the changes in the noise situation in individual measurement hours; the results are, however, systematically overestimated.

The differences between the prediction and measurement were also, in this case, assessed using Bland–Altman plots. The comparison of the levels *L*_Aeq,1h_ obtained by the prediction method Schall 03 and measurement is presented in Figure 8. The comparison of the levels *L*_Aeq,1h_ obtained by the prediction method MPVHD and measurement is presented in Figure 9. From Figure 8 and Figure 9, it is evident that the mean differences between the measured values and values predicted by the method Schall 03 are significantly smaller than with the values predicted by MPVHD. In the case of the MPVHD prediction method, the measured values of equivalent sound pressure level A are overestimated, whereas, for the Schall 03 method, the mean of predicted values is almost identical to the measured values.

Based on the train passes, noise maps of the surroundings of the measurement location for both days were created in the CadnaA program (according to the Schall 03 method valid in the Slovak Republic—Figure 10 and Figure 11).

These noise maps show that the place most exposed to noise is the central town part along the railway line. The color scale graded by 2 dB is illustrated on both maps. A on the studied location increased, and on the facades of the adjacent buildings, its value slightly exceeds the allowable value (60 dB) of the noise-determining quantity in terms of the valid legislation [25]. The studied part of the railway line is from one side (close to the measurement location) slightly sunken under the surrounding terrain. Buildings located on the other side of the railway line experience the highest noise exposure. Companies are located in these buildings. The building, opposite which the measurement location was situated, served as a seat of the otorhinolaryngology ward of the hospital in Zvolen. However, it was moved to another building within the hospital premises. It resulted in decreasing the noise which patients were exposed to. Subsequently, the despatching department was moved into the building, and, currently, the building is empty [26]. More trains during the second measurement campaign resulted in an increase in noise exposure of approx. 1 dB. The utilization of individual trains would need to be solved since some of the trains passing by were almost empty. Reducing the number of carriages associated with reducing the overall length of the train and, thus, the train pass-by time and overall noise exposure could be a solution. Although this part of the line is electrified, the electricity is used only to a certain extent to power the trains despite the fact that a detailed analysis had shown that the noise level produced by electric traction is significantly lower than that produced by diesel traction. A further issue is the poor condition of some trains or the railway line. Although limiting the train speed reduces the noise, it also increases the passing time.

Table 3 provides a comparison of measurement and prediction results from both measurement campaigns. Table 3 shows how much the given method overestimates (plus sign) or underestimates (minus sign) the in situ measurement results.

The differences are, in all cases, smaller than expanded measurement uncertainty. It is shown that it is difficult to determine the average speed during accelerating and decelerating of trains, whereby the speed affects the noise exposure to a great extent, and both methods are sensitive in this aspect. Both of these methods consider this parameter in a different way—Schall 03 uses logarithmic function and MPVHD exponential dependence. The prediction results are therefore relevant only when the speed limits at the given railway line part are observed. The Czech prediction method MPVHD does not consider the traction type, which also distorts the results. Both methods increase their inaccuracy in extreme situations, e.g., at a small number of trains passing by per a time unit or with noisier or more quiet trains which would correspond to the train type and speed. A further problem is the poor technical conditions of some trains. Similar results were also achieved by several other authors in studies that have tested the prediction models in local conditions. Implementing the Schall 03 method in the conditions of Serbian railways was discussed in the studies of Prascevic et al. (2013) [27]. Before applying the method at the national level, it was inevitable to validate and calibrate the method in accordance with the local conditions due to various technical and technological peculiarities of the railways. These processes were conducted at a railway line between the Belgrade and Romanian borders, providing for the calculation accuracy of noise indicators. The study results show that the calculated equivalent sound pressure level A is higher than the measured—for the reference time “day” the value was higher by 0.9 dB, and for the reference time “evening”, the value was higher by 0.6 dB. The results show that the equivalent sound pressure level A according to the prediction method Schall 03 is lower than the expanded measurement uncertainty; therefore, the method can be used for noise prediction at the railway line part in question. The major drawback of the method Schall 03 is the missing possibility of modifying the data describing the technical parameters of the machines, vehicles, other equipment, and railway infrastructure. In addition, the above-mentioned authors point to the fact that decreasing the train speed causes decreasing the calculated equivalent sound levels A; therefore, determining the speed is an important factor as well as determining the train length. Nonetheless, all these inaccuracies are much smaller than the expanded measurement uncertainty. Schall 03 method was developed for the specifics of the German railways; therefore, its implementation in Slovakia or Serbia requires creating sufficient corrections for the conditions of the trains, and types and quality of the railways in question. It is shown that the EU effort to unify the approach in all life spheres should not apply to the noise prediction, since the individual prediction methods were developed in specific conditions of the specific country, and they can work in different countries only if corrections are set in an appropriate way.

Lui et al. (2006) [28] conducted more extensive field measurements of a residential building. The objective of the study was to assess the noise emitted by trains passing over a bridge in the vicinity of the building in an urban environment. The measurement results (measured 1 m far from the building facade) were compared with results obtained from programs predicting noise from railway transport. It was shown that the most accurate results with the measurement were provided by the prediction model “Calculation of Railway Noise” (CRN) from Great Britain. Prediction method “Nordic Prediction Method for Train Noise” (NMT) also provided results at a good level, but the results are systematically overestimated (by 2–3 dB). Further prediction methods did not show results that would be accurate enough when compared with the measured values.

Nassiri et al. (2005) [29] attempted to design a relation for calculating the *L*_A,max_ (maximum sound pressure level) for the railway line Teheran–Karai for ordinary passenger trains. The form of the designed model was derived from the equation for predicting the *L*_A,max_ proposed in the guidelines by Harris Miller Miller and Hanson Inc. for the US Federal Transit Administration (FTA) [30] and in the French FTA prediction method for the railway noise. The prediction algorithm *L*_A,max_ was developed following 50 measurements in 5 locations, which were located 25, 35, 45, 55, and 65 m from the railway axis at a height of 1.5 m over the terrain. The reference distance was 25 m, and reference speed was 80 km/h. The study resulted in creating the relation for calculating the *L*_A,max_. The study also provided a background for further research and development of predicting and mapping the noise. Further noise studies of the above-mentioned team of authors were focused on monitoring the parameters of acceleration and deceleration of a train, as well as on predicting further parameters, e.g., SEL (sound exposure level—the level of a sound event relative to one second) or equivalent sound pressure level *L*_Aeq_, as well as on studying the relationship between them.

Pronello (2003) [31] analyzed the impact of railway transport on the environment. The study identified the variables significantly affecting the sound pressure level, defined the standard procedure for noise measurement, and developed a database for setting and calibration of noise models from railway transport. The pilot study deals with the noise at two railway lines passing through the town of Vercelli situated in the Northwest of Italy. The results indicated that, under certain conditions, variables normally affecting noise generation can be disregarded (e.g., at invariable environmental conditions, various train types do not significantly affect the sound pressure level). With diesel-powered trains with speeds lower than 70 km/h, a speed change of 30 to 40 km/h significantly affects the maximum sound pressure level (*L*_A,max_). In the case of electric trains with speeds lower than 80 km/h, a change of 20 to 30 km/h does not affect the *L*_A,max_ significantly. The noise pollution of diesel trains with low speed in the vicinity of a station is significantly affected by acceleration or deceleration. It was also shown that the site configuration and building situations affect the *L*_A,max_ significantly. The presence of high buildings along the railway line can increase the sound pressure level and can even suppress the advantages associated with the advancement in vehicle construction.

Currently, many authors have focused mainly on the noise generated by high-speed trains in densely populated areas [32,33,34]. With this type of trains, the aerodynamic noise is the dominant component.

## 4. Conclusions

The ever increasing noise intensity associated with human activities is becoming a more severe problem, mainly in developing urban agglomerations and their centers. Transport companies are also trying to improve their services in this aspect [35,36,37]. At present, potential synergistic effects of noise and other factors that may affect human health in the non-working and working environments are not taken into account [38]. Measurement results of the determining quantities and their comparison with prediction methods confirmed that the prediction methods describe the actual course of measured values with sufficient accuracy. The average difference between the measured and predicted values using the prediction method Schall 03 is smaller than in the case of the prediction method MPVHD. The use of the prediction method Schall 03 provided more values that differed only minimally from the measured values. Method Schall 03 provides, in this case, slightly better results when compared to the in situ measurements than the MPVHD method. Following the above-mentioned facts, it can be concluded that the use of the prediction method Schall 03 provides a more accurate value of *L*_Aeq,12h_ for the specific location. For Slovakia, prediction model Schall 03 appears to be more suitable, but national corrections for the train types and type of railway lines are very important. Therefore, it is important to test prediction models in various conditions of various states and implement national corrections into existing models or to refine and validate existing prediction models.

Predicting noise exposure from railway transport will also be one of the important tools for assessing and quantification of the noise burden of the population in the future. It is used not only for assessing the health risk of noise from railway transport, but it also provides information required for designing all types of measures for decreasing the harmful effect of this type of noise. With the development of urban agglomerations, the harmful impact of railway transport in such an environment will be discussed more and more. Additionally, in the future, the use of prediction methods will be a suitable tool for assessing the impact of noise from this type of transport [9].

Results of every measurement are valid for the given place, given time, and given weather conditions. Several measurements were carried out at several sites and several times by us and other organizations. The paper, however, presents a case study of the selected location. Nevertheless, the conclusions are also in accordance with other measurements. The impact of individual parameters affecting the prediction models and the value of the determining *L_Aeq,1h_* was established, and the strengths and weaknesses of the used models were analyzed following their statistical evaluation (speed and continuity, traction and braking, extreme situations, the technical condition of trains).

## Figures and Tables

**Figure 1 ijerph-17-03616-f001:**
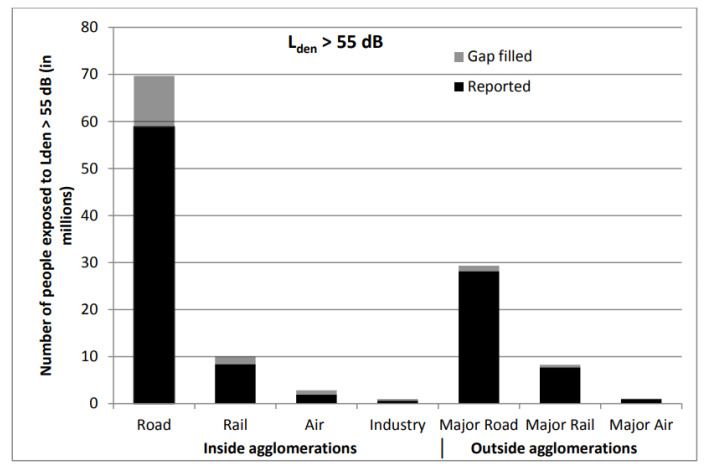
The number of people exposed to environmental noise in Europe *L*_den_ > 55 dB in 28 EU member states in 2012, in and outside the urban agglomerations [6]. Note: Gap filing has been carried out based on the methodology published in “Noise in Europe 2014”, available at http://www.eea.europa.eu/publications/noise-in-europe-2014.

**Figure 2 ijerph-17-03616-f002:**
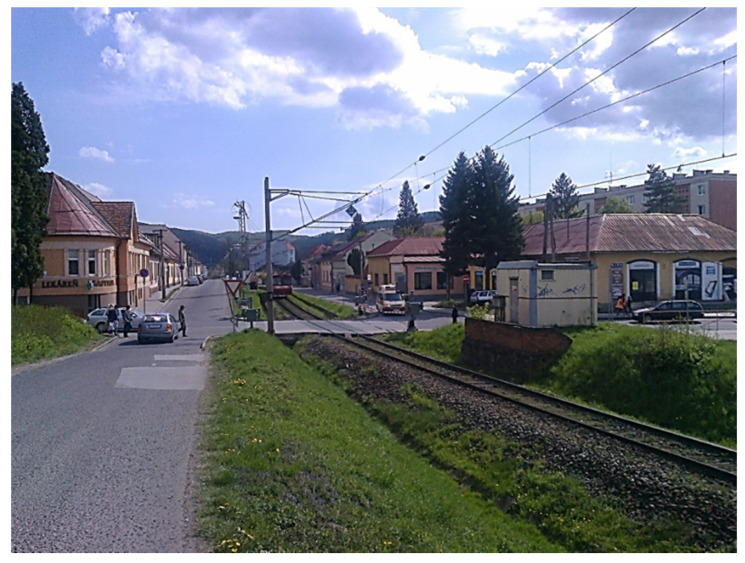
Buildings in the vicinity of the assessed location ZV1/ZV2 Zvolen.

**Figure 3 ijerph-17-03616-f003:**
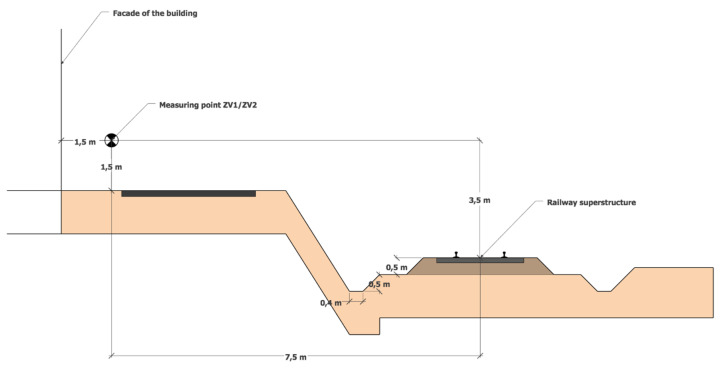
Measurement location ZV1/ZV2 with regard to the railway line.

**Figure 4 ijerph-17-03616-f004:**
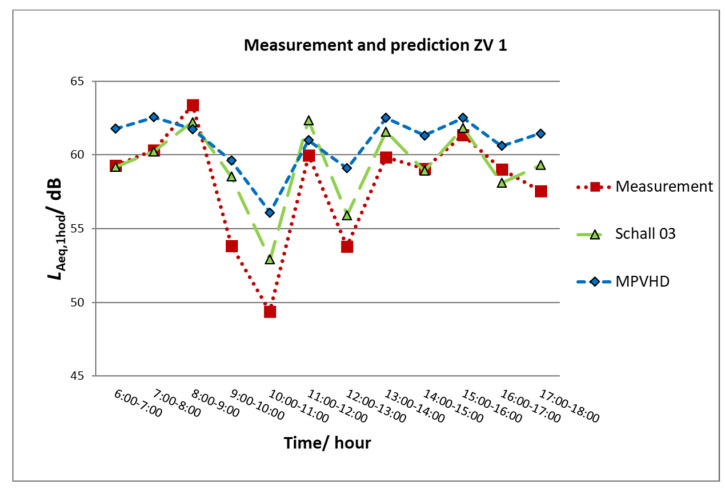
Graphical comparisons of the prediction and measurement of ZV1.

**Figure 5 ijerph-17-03616-f005:**
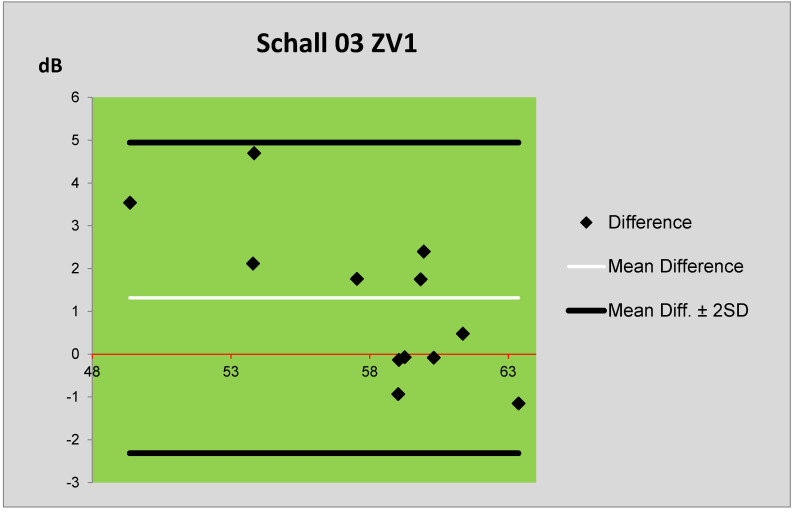
Bland–Altman plot of differences between the prediction (Schall 03) and measurement of ZV1.

**Figure 6 ijerph-17-03616-f006:**
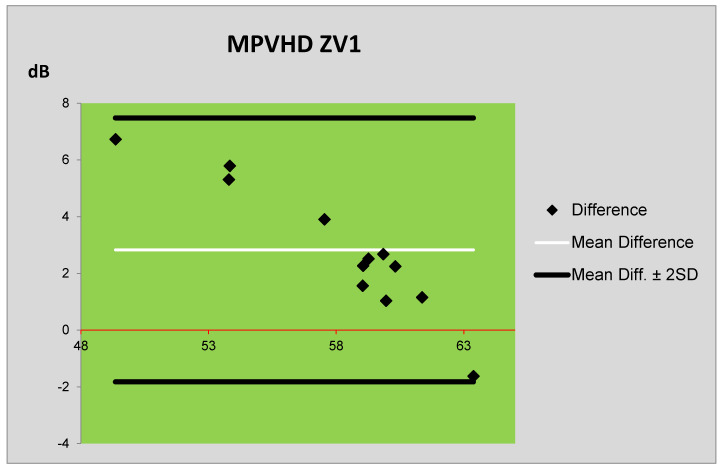
Bland–Altman plot of differences between the prediction (MPVHD) and measurement of ZV1.

**Figure 7 ijerph-17-03616-f007:**
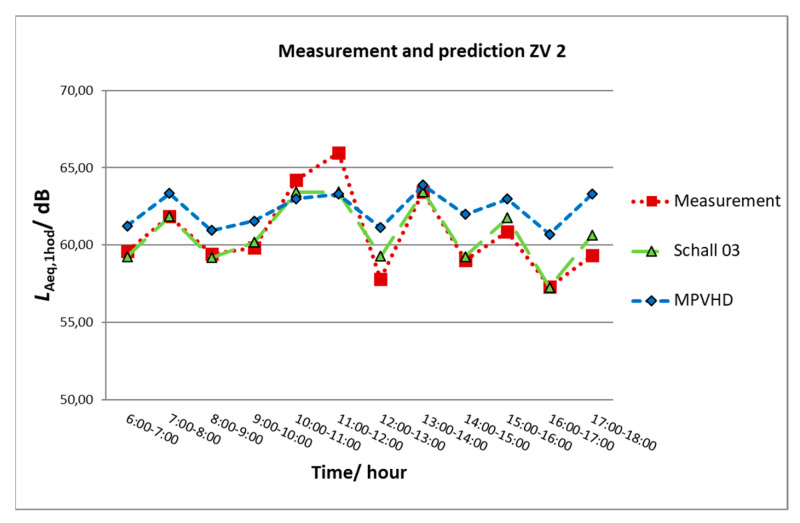
Graphical comparison of prediction and measurement of ZV2.

**Figure 8 ijerph-17-03616-f008:**
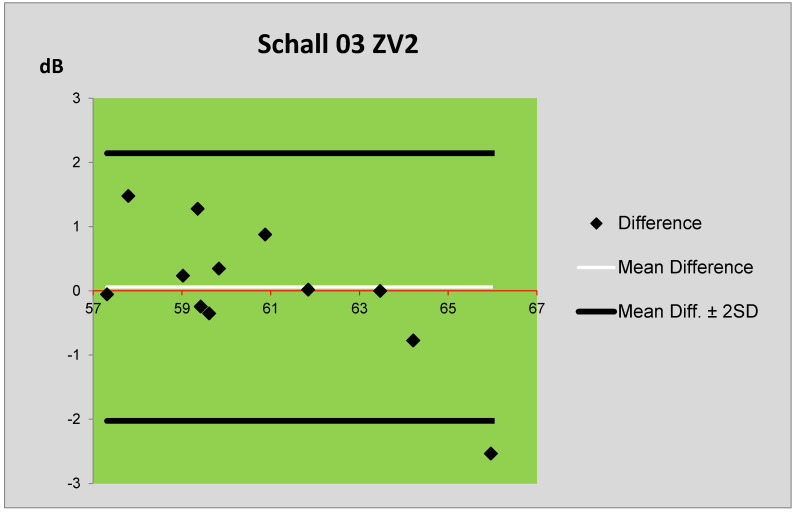
Bland–Altman plot of differences between the prediction (Schall 03) and measurement of ZV2.

**Figure 9 ijerph-17-03616-f009:**
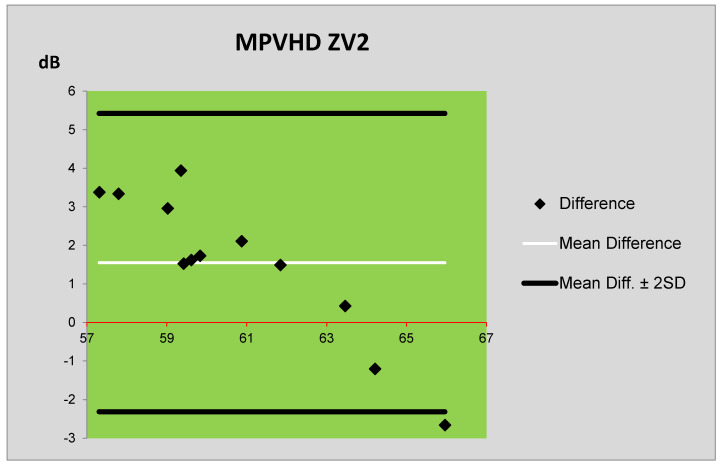
Plot of differences between the prediction (MPVHD) and measurement of ZV2.

**Figure 10 ijerph-17-03616-f010:**
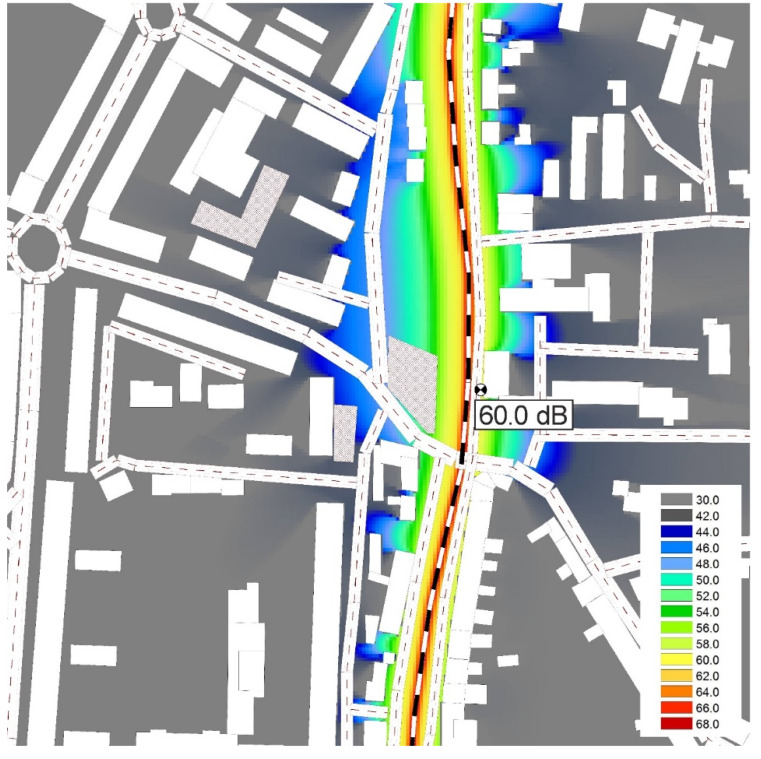
Noise map ZV1.

**Figure 11 ijerph-17-03616-f011:**
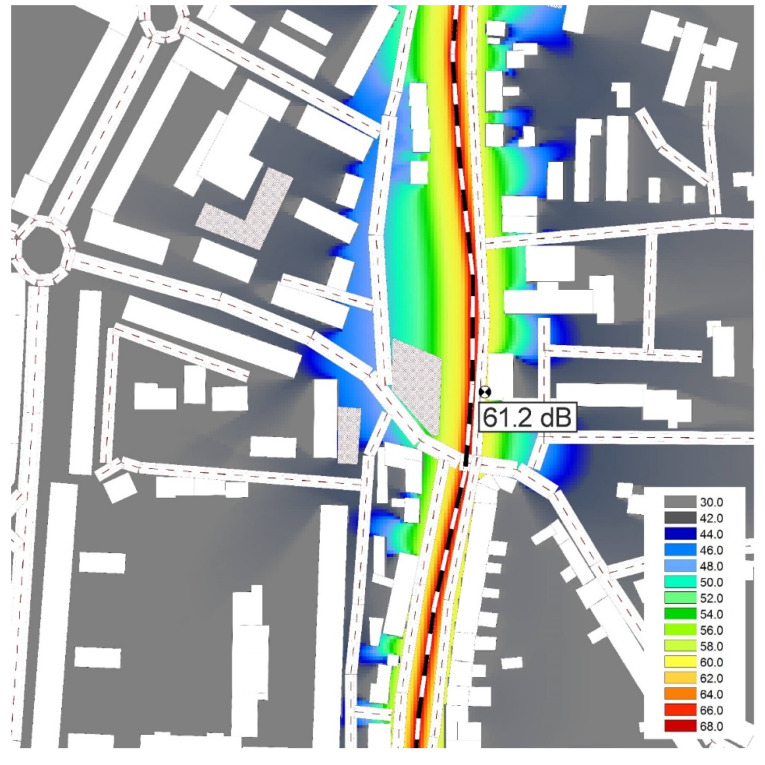
Noise Map ZV2.

**Table 1 ijerph-17-03616-t001:** Number and types of trains during the measurement of ZV1.

Direction Zvolen–Banská Bystrica		Direction Banská Bystrica–Zvolen	
Type	Number	Type	Number
Os	11	Os	15
Zr	1	Zr	2
R	5	R	2
N	1	N	1
Vú	3	Vú	0
Total	21	Total	20

(Os—ordinary passenger train, Zr—semi-fast regional train, R—ordinary fast train, N—goods train, Vú–maintenance train).

**Table 2 ijerph-17-03616-t002:** Number and types of trains during the measurement of ZV2.

Direction Zvolen–Banská Bystrica		Direction Banská Bystrica–Zvolen	
Type	Number	Type	Number
Os	15	Os	16
Zr	0	Zr	0
R	6	R	6
N	2	N	2
Vú	3	Vú	0
total	26	total	24

(Os—ordinary passenger train, Zr—semi-fast regional train, R—ordinary fast train, N—goods train, Vú—maintenance train).

**Table 3 ijerph-17-03616-t003:** Comparison of the measurement and prediction results.

Measurement	Schall 03	MPVHD
ZV1	+0.6 dB	+1.3 dB
ZV2	−0.3 dB	+0.9 dB

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
