# Peer review of "Measurement and Prediction of Railway Noise Case Study from Slovakia"

_ijerph, 2020, doi:10.3390/ijerph17103616_

Round 1

Reviewer 1 Report

The work is more like a research report than a scientific article. A broad description of calculation methods is missing. Only assumptions and results can be seen.

The results differ from each other and this is a very important observation. However, in the summary there is a lack of a clear explanation of the reasons and further recommendations.

In the particular case which the authors had to deal with, the results were as presented. Maybe it would be worth pointing out what are the risks if you try to extrapolate the obtained results (from this single measurement point) into the entire railway network. What additional research should be carried out? What positions of the measurement points should be chosen? Maybe double-track lines with a large number of high-speed electric trains, maybe a freight line? In each case, the result may be completely different and it is worth mentioning in the summary.

Author Response

Review 1

Dear Reviewer,

We kindly thank you for the opportunity given to us to revising our work in order to reach the high standards or this Journal.

We greatly appreciate thoughtful comments that helped improve the manuscript. We trust that all of your comments have been addressed accordingly in a revised manuscript. Below are our point-by-point responses to your comments. The appropriate corrections and modifications to the revised manuscript are presented with different highlighted font each time. We hope that our appropriate corrections based on your comments increased the quality of our work.

A broad description of calculation methods is missing.

The Deutsche Bundesbahn developed and use the Schall 03 2006 prediction method. It is used for predicting the railway noise and creating railway noise maps in Slovakia. The equivalent sound pressure level A is in this case called the emission level Lm,E and is determined I the perpendicular distance of 25 m from the railway line axis at the height of 4 m above the terrain. Emission levels Lm,E are determined according to the equation :

Lm,E= 10.log[Σ10(0,1.(51+DFZ+DD+DL+DV))]+DTt+DBr+DLc+DRa

Where DFz, DD, DL, Ds are corrections for the specific train and DTt, DBr, DLc, DRa are corrections characterising the railway line. More detailed explanation of the methodology is Mohler a kol., 2014.

The current legislation in the Czech Republic issued in 1991 and amended in 1996 is called the Methodical instructions for the calculation of sound pressure level from transport (MPVHD).

The equivalent sound pressure level A in dB is established for the assessed site located in the distance of 7.5 m from the railway line axis. The equation is as follows:

Y = 10.log X + 40                                                                                          

The value X (in dB as well) is calculated according to the equation:

X = 140. F4. F5. F6. m                                          

Where F4 is the factor characterising the impact of traction, F5 is the factor of instantaneous speed at the given point of the railway line and F6 is the factor considering the average total number of vehicles. The methodology is further explained in Liberko, 1996.

However, in the summary there is a lack of a clear explanation of the reasons and further recommendations.

The conclusions were modified accordingly.

In the particular case which the authors had to deal with, the results were as presented. Maybe it would be worth pointing out what are the risks if you try to extrapolate the obtained results (from this single measurement point) into the entire railway network. What additional research should be carried out? What positions of the measurement points should be chosen? Maybe double-track lines with a large number of high-speed electric trains, maybe a freight line? In each case, the result may be completely different and it is worth mentioning in the summary.

One measurement cannot be used for generalising, of course. Results of every measurement is valid for the given place, given time and given weather conditions. Several measurements were carried out at several sites and several times by us and other organizations. The paper, however, presents a case study of the selected location. Nevertheless, the conclusions are in accordance also with other measurements.

Measurements that are used to create noise maps cover a broader area. The present study focuses on a case study of a smaller area in question and describes in more detail a selected area affected by higher noise levels from railway transport.

http://www.hlukovamapa.sk/shm_zsr_2016.html

Reviewer 2 Report

Dear Authors,

    there are (at least) some improvements I will suggest for your pubblication:

  1. First of all, I would suggest to insert more details concerning both the two calculation methods you have considered and the measurements procedure you have followed;
  2. Concerning the experimental work, what do you mean saying "The residual noise from the surrounding sources (mainly road transport) is during the measurement sufficiently decreased" (line 87)? Did you evaluate the noise of each event (passage) using the corresponding SEL?
  3. Please, specify what you want to demonstrate through Table n. 3, change its caption and clarify that ZV1 and ZV2 refer to the noise maps in Figures 9 and 10;
  4. Figure 4 and figure 5: please use the same scale. Moreover, it is not clear the difference between the mean line and the mean +/- two standard deviation: please use a dashed line for the mean, instead of different thicknesses;
  5. Figure 7 and 8: same as previous point;  
  6. Figure 9 and Figure 10: what would it happen turning on road noise? And what would be the noise map considering road noise only? Evaluating the contribution to environmental noise, instead of railroad source emission, could be meaningful as well.
  7. Figure 9 and 10: please, specify which CadnaA parameters have been employed; did you consider building reflections?

Author Response

Review 2

Dear Reviewer,

We kindly thank you for the opportunity given to us to revising our work in order to reach the high standards or this Journal.

We greatly appreciate thoughtful comments that helped improve the manuscript. We trust that all of your comments have been addressed accordingly in a revised manuscript. Below are our point-by-point responses to your comments. The appropriate corrections and modifications to the revised manuscript are presented with different highlighted font each time. We hope that our appropriate corrections based on your comments increased the quality of our work.

First of all, I would suggest to insert more details concerning both the two calculation methods you have considered and the measurements procedure you have followed;

The Deutsche Bundesbahn developed and use the Schall 03 2006 prediction method. It is used for predicting the railway noise and creating railway noise maps in Slovakia. The equivalent sound pressure level A is in this case called the emission level Lm,E and is determined I the perpendicular distance of 25 m from the railway line axis at the height of 4 m above the terrain. Emission levels Lm,E are determined according to the equation  

Lm,E= 10.log[Σ10(0,1.(51+DFZ+DD+DL+DV))]+DTt+DBr+DLc+DRa                 

Where DFz, DD, DL, Ds are corrections for the specific train and DTt, DBr, DLc, DRa are corrections characterising the railway line. More detailed explanation of the methodology is Mohler a kol., 2014.

The current legislation in the Czech Republic issued in 1991 and amended in 1996 is called the Methodical instructions for the calculation of sound pressure level from transport (MPVHD).

The equivalent sound pressure level A in dB is established for the assessed site located in the distance of 7.5 m from the railway line axis. The equation is as follows:

 Y = 10.log X + 40                                                                                                                              

The value X (in dB as well) is calculated according to the equation:

X = 140. F4. F5. F6. m                                                                               

Where F4 is the factor characterising the impact of traction, F5 is the factor of instantaneous speed at the given point of the railway line and F6 is the factor considering the average total number of vehicles. The methodology is further explained in Liberko, 1996.

Concerning the experimental work, what do you mean saying "The residual noise from the surrounding sources (mainly road transport) is during the measurement sufficiently decreased" (line 87)? Did you evaluate the noise of each event (passage) using the corresponding SEL?

When it is not practical to measure the equivalent sound pressure level Leq for the required number of sound events, the level LE is measured for every single sound event. A minimum number of sound events of the operated sound sources is measured as specified in article 6. Every sound event during a time period, which is sufficiently long for recording all important noise contributions, is measured. With sound events, the measurement is carried out until the sound pressure level decreases by minimum 10 dB under the maximum level.

Residual noise was determined as a maximum measured value LAF95max = 52.2 dB during the entire measurement. The lowest value of the A level of the sound exposure determined at passing by of individual trains was LAE = 64.7 dB (Tab. 10). Since the difference between these two values is more than 10 dB, all measured values of sound pressure levels are valid for all train passes by. The conducted measurement thus does not have to be corrected using the residual noise correction.

Please, specify what you want to demonstrate through Table n. 3, change its caption and clarify that ZV1 and ZV2 refer to the noise maps in Figures 9 and 10;

Table description corrected

Figure 4 and figure 5: please use the same scale. Moreover, it is not clear the difference between the mean line and the mean +/- two standard deviation: please use a dashed line for the mean, instead of different thicknesses;

The line colour was changed. The scale remained unchanged; the scale must be always adjusted to the displayed values. If the results were close to the average, they would not be displayed with a uniform scale.

Figure 7 and 8: same as previous point; 

The line colour was changed. The scale remained unchanged; the scale must be always adjusted to the displayed values. If the results were close to the average, they would not be displayed with a uniform scale.

Figure 9 and Figure 10: what would it happen turning on road noise? And what would be the noise map considering road noise only? Evaluating the contribution to environmental noise, instead of railroad source emission, could be meaningful as well.

This is beyond the scope of the study. We agree that it is necessary to quantify the noise from the road traffic and environmental noise in order to describe the overall noise situation. However, it is a complex topic going beyond the present paper.

Figure 9 and 10: please, specify which CadnaA parameters have been employed; did you consider building reflections?

When creating the model, main parameters (e.g. number of passes by, the train lengths, the train type and speed, the height of the track bed etc.) were used. The dimensions and heights of the surrounding buildings were defined according to the internet map client ZBGIS.

The grid calculation with a grid spacing of 1 m was set and the height of horizontal grid was 1.5 m. Further parameters were predefined by the CadnA programme when the calculation was input according to the German prediction model Schall 03. A more detailed assessment and setting of parameters will be discussed in further research studies.

In the research we did not consider the building reflection due to simplifying the final model, when this requirement was not significant for us.

Reviewer 3 Report

Despite six authors and 37 references, this content is a rather weak piece of work. The questioning as well as the research are very faint, and the work hardly adds anything to the state of knowledge. The submission is therefore rather weak and is of only local advisory significance, the whole is too weak for an international, let alone scientific publication.

For these reasons, this reviewer recommends rejection of this submission.

If the editor would conclude otherwise, this reviewer adds a series of possibly helpful comments and suggestions.

More detailed remarks and comments

A proper review of the English-language wording is necessary.

The abstract already illustrates the very limited ambitions.

Line 56 & Page 2 ‘sprawls’ why not ‘agglomerations’ as on figure 1?

Explain ‘gap filled’ in figure 1.

Line 82 ‘It is a two-channel analyser’ was the two-channel characteristic of any use?

Line 88 ‘The residual noise from the surrounding source… sufficiently decreased…also during the actual measurements.’ Specify your criterion.

Line 90 ‘The measurements and predictions were carried out repeatedly (on 3 October 2011

and on 24 April 2015)’ In terms of chronology, this is quite long time ago: this does not indicate scientific nor practical urgency…

Line 97 ‘After the calibration, the sound pressure analyser better sound level meter was placed at the measurement location …. The embankment is 0.5 m high.’ For the comfort of the reader it would be more  efficient to provide the section’s  drawing!

Line 118 ‘is illustrated using the noise maps, which were created using the CadnaA’ although colourful illustrations, these plots are not really relevant for the subject of the submission.

Page  9 superficial page filling explanation 208-226-264. Should be condensed!

Line 274-303 we should find this part at the beginning of the submission: the state of the art should be discussed at the start, not at the end.

Line 304 4. Conclusions: stick to the real conclusions of the work!

Line 305-311 is too general and should at this position in the text, better not refer to bibliographic references!

Author Response

Review 3

Dear Reviewer,

We kindly thank you for the opportunity given to us to revising our work in order to reach the high standards or this Journal.

We greatly appreciate thoughtful comments that helped improve the manuscript. We trust that all of your comments have been addressed accordingly in a revised manuscript. Below are our point-by-point responses to your comments. The appropriate corrections and modifications to the revised manuscript are presented with different highlighted font each time. We hope that our appropriate corrections based on your comments increased the quality of our work.

Line 56 & Page 2 ‘sprawls’ why not ‘agglomerations’ as on figure 1?

The expression 'sprawls' was  replaced by the expression 'agglomerations'.

Explain ‘gap filled’ in figure 1.

 * Gap filing has been carried out based on the methodology published in "Noise in Europe 2014", available at http://www.eea.europa.eu/publications/noise-in-europe-2014

Line 82 ‘It is a two-channel analyser’ was the two-channel characteristic of any use?

The description was taken over from the manufacturer’s description; it was removed from the text..

Line 88 ‘The residual noise from the surrounding source… sufficiently decreased…also during the actual measurements.’ Specify your criterion.

When it is not practical to measure the equivalent sound pressure level Leq for the required number of sound events, the level LE is measured for every single sound event. A minimum number of sound events of the operated sound sources is measured as specified in article 6. Every sound event during a time period, which is sufficiently long for recording all important noise contributions, is measured. With sound events, the measurement is carried out until the sound pressure level decreases by minimum 10 dB under the maximum level.

Residual noise was determined as a maximum measured value LAF95max = 52.2 dB during the entire measurement. The lowest value of the A level of the sound exposure determined at passing by of individual trains was LAE = 64.7 dB (Tab. 10). Since the difference between these two values is more than 10 dB, all measured values of sound pressure levels are valid for all train passes by. The conducted measurement thus does not have to be corrected using the residual noise correction.

Line 90 ‘The measurements and predictions were carried out repeatedly (on 3 October 2011

and on 24 April 2015)’ In terms of chronology, this is quite long time ago: this does not indicate scientific nor practical urgency…

One measurement cannot be used for generalising, of course. Results of every measurement is valid for the given place, given time and given weather conditions. Several measurements were carried out at several sites and several times by us and other organizations. The paper, however, presents a case study of the selected location. Nevertheless, the conclusions are in accordance also with other measurements.

Measurements that are used to create noise maps cover a broader area. The present study focuses on a case study of a smaller area in question and describes in more detail a selected area affected by higher noise levels from railway transport.

http://www.hlukovamapa.sk/shm_zsr_2016.html

Line 97 ‘After the calibration, the sound pressure analyser better sound level meter was placed at the measurement location …. The embankment is 0.5 m high.’ For the comfort of the reader it would be more  efficient to provide the section’s  drawing!

A drawing has been added

Line 118 ‘is illustrated using the noise maps, which were created using the CadnaA’ although colourful illustrations, these plots are not really relevant for the subject of the submission.

The CadnA software works with the prediction methodology Schall 03, which was tested in the study, and therefore, the graphical output of the noise map of the area in question is considered relevant for supplementing this case study.

Page  9 superficial page filling explanation 208-226-264. Should be condensed!

Results and conclusions were modified.

Line 274-303 we should find this part at the beginning of the submission: the state of the art should be discussed at the start, not at the end.

It is comparison of results not the state of the art.

Line 304 4. Conclusions: stick to the real conclusions of the work!

Results and conclusions were modified.

Line 305-311 is too general and should at this position in the text, better not refer to bibliographic references!

Results and conclusions were modified.

Reviewer 4 Report

General comments

The measurement procedure follows the ISO 1996-2:2017 standard, which is a rather general document. In determining your experimental procedure, have you also considered the type-testing standard EN ISO 3095?

Schall-03 is a well-known prediction model (I am not familiar with MPVHD), but it is rather outdated: have you considered making comparisons with the EC harmonized CNOSSOS-EU method?

No data are mentioned in the paper about background noise due to traffic and/or stationary sources: can you comment on this aspect? Could background noise justify the systematic overestimate of predicted values in the measurement campaign ZV1, also considering that predictions seem to improve in measurement campaign ZV2, in which a higher number of train passages occurred?

Apparently, train speed during pass-by was not measured. How did you handle this information, which is required by the prediction methods among the inputs?

At lines 208-211 reference is made to allowable noise according to valid legislation: I suggest adding the relevant article of law to the references.

The literature review at lines 233-303 contains some useful remarks but looks somewhat disconnected from the previous discussion: please, try to improve the connection between the two parts.

Specific comments

Please check the use of the term “urban sprawl”, which is used extensively throughout the article and sounds rather awkward to me. Do you mean “agglomeration”, as defined in the European Noise Directive?

Line 25 (and elsewhere): Shall 03 -> Schall 03

Lines 66-69: it is clear what you mean, but the text should be rephrased

Line 121: “At the time of the first measurement…” -> “At the time of the first measurement campaign (ZV1)…”

Line 128 (and elsewhere): please round all figures in dB to the first decimal (e.g. 59.27 dB -> 59.3 dB)

Line 138 (and elsewhere): “equivalent sound level A” do you mean “A-weighted equivalent continuous sound pressure level”? If so, please use the latter form, as specified by technical standards

Line 154: “Schall 03 method provided more values, which are different only minimally….” -> “Schall 03 method provided more values that are only minimally different ….”

Line 167: “The repeated measurement…” -> “The repeated measurement campaign (ZV2)…” or “The second measurement campaign (ZV2)…”

Line 179: What do you mean by “determining quantities”?

Lines 222-223: “the time needed for passing” -> “the train pass-by time”

Line 269: “the most similar results” -> “the most accurate results”

Line 287-288: “He identified” -> “She identified” (C. Pronello is a woman)

Line 291: “Varcelli” -> “Vercelli”

Author Response

Review 4

Dear Reviewer,

We kindly thank you for the opportunity given to us to revising our work in order to reach the high standards or this Journal.

We greatly appreciate thoughtful comments that helped improve the manuscript. We trust that all of your comments have been addressed accordingly in a revised manuscript. Below are our point-by-point responses to your comments. The appropriate corrections and modifications to the revised manuscript are presented with different highlighted font each time. We hope that our appropriate corrections based on your comments increased the quality of our work.

The measurement procedure follows the ISO 1996-2:2017 standard, which is a rather general document. In determining your experimental procedure, have you also considered the type-testing standard EN ISO 3095?

EN ISO 3095 – This international standard specifies measurement methods and conditions to obtain reproducible and comparable exterior noise emission levels and spectra for all kinds of vehicles operating on rails or other types of fixed track. The annexes specify conditions for measurement of the exhaust noise and intake noise, measurement on stationary vehicles or during vehicle acceleration, measurement of noise at platforms, bridges, viaducts and in tunnels. Thus, this standard does not specify the conditions for noise measurement from the railway transport in the actual transport of people and goods.

Schall-03 is a well-known prediction model (I am not familiar with MPVHD), but it is rather outdated: have you considered making comparisons with the EC harmonized CNOSSOS-EU method?

EC harmonized CNOSSOS-EU method is based on the Schall 03 methodology, what can be seen in the references. It is a general noise prediction methodology with a unified nomenclature with the aim of obtaining noise prediction methodology from the railway transport for the whole Europe. This method does not reflect the condition of the vehicles, trackbed or the technical condition of the vehicles in individual countries. In the Slovak Republic the modified methodology Schall 03 is legislatively anchored and used; therefore, it was used also in the study.

No data are mentioned in the paper about background noise due to traffic and/or stationary sources: can you comment on this aspect? Could background noise justify the systematic overestimate of predicted values in the measurement campaign ZV1, also considering that predictions seem to improve in measurement campaign ZV2, in which a higher number of train passages occurred?

When it is not practical to measure the equivalent sound pressure level Leq for the required number of sound events, the level LE is measured for every single sound event. A minimum number of sound events of the operated sound sources is measured as specified in article 6. Every sound event during a time period, which is sufficiently long for recording all important noise contributions, is measured. With sound events, the measurement is carried out until the sound pressure level decreases by minimum 10 dB under the maximum level.

Residual noise was determined as a maximum measured value LAF95max = 52.2 dB during the entire measurement. The lowest value of the A level of the sound exposure determined at passing by of individual trains was LAE = 64.7 dB (Tab. 10). Since the difference between these two values is more than 10 dB, all measured values of sound pressure levels are valid for all train passes by. The conducted measurement thus does not have to be corrected using the residual noise correction.

Apparently, train speed during pass-by was not measured. How did you handle this information, which is required by the prediction methods among the inputs?

The speed during pass-by was measured with every train.

At lines 208-211 reference is made to allowable noise according to valid legislation: I suggest adding the relevant article of law to the references.

Reference to legislation has been added.

The literature review at lines 233-303 contains some useful remarks but looks somewhat disconnected from the previous discussion: please, try to improve the connection between the two parts.

The connection has been added.

Please check the use of the term “urban sprawl”, which is used extensively throughout the article and sounds rather awkward to me. Do you mean “agglomeration”, as defined in the European Noise Directive?

The expression 'sprawls' was replaced by the expression 'agglomerations'.

Line 25 (and elsewhere): Shall 03 -> Schall 03

The error has been fixed.

Lines 66-69: it is clear what you mean, but the text should be rephrased

The text was changed.

Line 121: “At the time of the first measurement…” -> “At the time of the first measurement campaign (ZV1)…”

This has been clarified.

The measurements ZV1 and ZV2 have been explained in the text.

Line 128 (and elsewhere): please round all figures in dB to the first decimal (e.g. 59.27 dB -> 59.3 dB)

Data were rounded.

Line 138 (and elsewhere): “equivalent sound level A” do you mean reference to legislation has been added equivalent continuous sound pressure level”? If so, please use the latter form, as specified by technical standards

It has been corrected.

Corrected to “equivalent sound level pressure A

Line 154: “Schall 03 method provided more values, which are different only minimally….” -> “Schall 03 method provided more values that are only minimally different ….”

The sentence has been modified.

Line 167: “The repeated measurement…” -> “The repeated measurement campaign (ZV2)…” or “The second measurement campaign (ZV2)…”

The issue has been corrected.

Line 179: What do you mean by “determining quantities”?

The quantity describing a phenomenon and which is used to carry out assessment and comparison.

Lines 222-223: “the time needed for passing” -> “the train pass-by time”

The error has been fixed.

Line 269: “the most similar results” -> “the most accurate results”

The error has been fixed.

Line 287-288: “He identified” -> “She identified” (C. Pronello is a woman)

The error has been fixed.

Line 291: “Varcelli” -> “Vercelli”

The error has been fixed.

Round 2

Reviewer 3 Report

Taking into account the improvements and additions made and also taking into account the mild assessment by the fellow reviewer, I have come to a positive assessment of this manuscript.
The changes made in response to the comments and suggestions have improved the editorial quality of the text, but this submission remains relatively limited in terms of content and research design.

Author Response

Dear Reviewer,

Thank you for your assessment of our manuscript. Although you consider its quality relatively limited, thank you for your comments, which have significantly helped to improve its quality.